# Seed Geometry in the Vitaceae

**DOI:** 10.3390/plants10081695

**Published:** 2021-08-18

**Authors:** Emilio Cervantes, José Javier Martín-Gómez, Diego Gutiérrez del Pozo, Ángel Tocino

**Affiliations:** 1Instituto de Recursos Naturales y Agrobiología del Consejo Superior de Investigaciones Científicas (IRNASA-CSIC), Cordel de Merinas, 40, 37008 Salamanca, Spain; jjavier.martin@irnasa.csic.es; 2Departamento de Conservación y Manejo de Vida Silvestre (CYMVIS), Universidad Estatal Amazónica (UEA), Carretera Tena a Puyo Km, 44, Puyo 150950, Ecuador; diego.gutierrez.pozo@gmail.com; 3Departamento de Matemáticas, Facultad de Ciencias, Universidad de Salamanca, Plaza de la Merced 1-4, 37008 Salamanca, Spain; bacon@usal.es

**Keywords:** endosperm, geometry, morphology, seed shape, Vitaceae

## Abstract

The Vitaceae Juss., in the basal lineages of Rosids, contains sixteen genera and 950 species, mainly of tropical lianas. The family has been divided in five tribes: Ampelopsideae, Cisseae, Cayratieae, Parthenocisseae and Viteae. Seed shape is variable in this family. Based on new models derived from equations representing heart and water drop curves, we describe seed shape in species of the Vitaceae. According to their similarity to geometric models, the seeds of the Vitaceae have been classified in ten groups. Three of them correspond to models before described and shared with the Arecaceae (lenses, superellipses and elongated water drops), while in the seven groups remaining, four correspond to general models (waterdrops, heart curves, elongated heart curves and other elongated models) and three adjust to the silhouettes of seeds in particular genera (heart curves of *Cayratia* and *Pseudocayratia*, heart curves of the Squared Heart Curve (SqHC) type of *Ampelocissus* and *Ampelopsis* and Elongated Superellipse-Heart Curves (ESHCs), frequent in *Tetrastigma* species and observed also in *Cissus* species and *Rhoicissus rhomboidea*). The utilities of the application of geometric models for seed description and shape quantification in this family are discussed.

## 1. Introduction

The Vitaceae Juss. contains sixteen genera with ca. 950 species of lianas primarily distributed in the tropics with some genera in the temperate regions. The Leeaceae Dumort., with a single genus of 34 species, mostly shrubs and small trees rather than lianas, included in the family in the APG IV [1], was later recognized as a separate family [2,3]. Both families constitute the order Vitales, one of the basal lineages of Rosids, whose closest relative remains controversial [4,5].

The Vitaceae has been divided in five tribes [3] (Table 1): (I) Ampelopsideae J. Wen and Z. L. Nie (*Ampelopsis* Michx., *Nekemias* Raf., *Rhoicissus* Planch., *Clematicissus* Planch.); (II) Cisseae Rchb. (*Cissus* L.); (III) Cayratieae J.Wen and L.M.Lu (*Cayratia* Juss. Ex Guill, *Causonis* Raf., *Acareosperma* Gagnep., *Afrocayratia*, *Cyphostemma* (Planch.) Alston, *Pseudocayratia* J.Wen, L.M.Lu and Z.D.Chen, *Tetrastigma* Planch.); (IV) Parthenocisseae J.Wen and Z.D.Chen (*Parthenocissus* Planch.) and (V) Viteae Dumort (*Ampelocissus* Planch., and *Vitis* L.). *Cayratia* and *Cyphostemma* were included in *Cissus* by Linné and Planchon considered the former as a section of *Cissus* [6,7].

*Cissus* is the largest genus in the family with 300 species of complex classification [8]. *Cyphostemma* is second, with 200 species of an interesting diversity in their range of distribution as well as in growth habits (vines and lianas, herbs, stem succulents and a tree) [9]. *Vitis* has seventy-five inter-fertile wild species distributed in three continents under subtropical, Mediterranean and continental-temperate climatic conditions. *Vitis vinifera* L. is the species with highest economic importance in the family with some taxonomic uncertainty about the differentiation between *V. vinifera* L. subsp. *vinifera* and *V. vinifera* L. subsp. *sylvestris* (Willd.) Hegi [10,11]. Thousands of cultivars of *V. vinifera* are used worldwide in Viticulture. Species of other genera are widely cultivated, such as *Parthenocissus quinquefolia* (L.) Planch., the Virginia creeper, in temperate areas, and *Cissus incisa* Des Moul., the grape ivy, in tropical areas. Species of the genus *Tetrastigma* are the only host plants for the parasitic plant *Rafflesia arnoldii* R.Br., Rafflesiaceae, which is native only to a few areas within the Malay Archipelago [12].

The taxonomy and phylogenetic relationships of the Vitaceae are far from complete and will benefit from an accurate description of seeds in the unambiguous terms of geometry. From a practical point of view, the classification based on geometric models may contribute to the distinction between wild and crop grapes of *Vitis vinifera* [11].

The main objective of this review is to provide a framework for the description of seed morphology in the Vitaceae based on geometric models. A recent review of this subject in the Arecaceae described morphological types in the seeds of this family based on the similarity of seed images to geometric figures, like ellipses, ovals and others [13]. Members of both families, the Arecaceae and the Vitaceae, were present in the Neotropical flora in the Eocene [14], and the description of their seeds may serve as a model to develop this work in other plant families.

## 2. Seed Morphology in the Vitaceae

### 2.1. Quantification of Seed Shape by Geometric Models

The silhouettes of bi-dimensional images of seeds often resemble geometric figures that can be used as models for the description and quantification of seed shape in plant families. A recent review of the geometry of seeds in the Arecaceae described a series of models useful for the analysis of seed shape in this family [13]. Geometric models included the ellipses (the circle is a particular type of ellipse), ovals, lemniscates, superellipses, cardioid and derivatives, lenses and the water drop curve [13]. The reader is referred to this review for the algebraic description of the models and their application in the morphometry of seeds in the Arecaceae. The application of geometric models in morphometry is based on the comparison of bi-dimensional images of well oriented seeds with these figures by means of image programs working in two layers (Adobe Photoshop, Corel Photo Paint…). The two images (seed and model) can be superimposed searching for a maximum similarity and the ratio between shared and total surface areas, that we have termed *J* index, is calculated with the data obtained in ImageJ [15]. *J* index measures the percent of similarity between two images (the seed and the model) and provides information on overall seed shape [16,17]. Bidimensional seed images of many plant species adjust well to one of three morphological types: the ellipse, the oval and the cardioid [18]. The seeds of the model plant *Arabidopsis thaliana* (L.) Heynh., those of the model legumes *Lotus japonicus* (Regel) K.Larsen and *Medicago truncatula* Gaertn., as well as the seeds of *Capparis spinosa* L., in the Capparaceae and *Rhus tripartita* DC. in the Anacardiaceae adjust well to cardioids or modified cardioids [19,20,21,22,23]. The seeds of *Ricinus communis* L. and *Jatropha curcas* L. in the Euphorbiaceae and those of cultivars of *Triticum* sp. in the Poaceae adjust well to ellipses of varied x/y ratio [24,25,26]. Oval shaped seeds occur frequently in the Cucurbitaceae, Berberidaceae, Eupteleaceae and Lardizabalaceae [27,28], while the cardioid is more common in Papaveraceae [28]. A given geometric type is sometimes associated with other morphological or ecological characteristics. For example, cardioid-type seeds were observed to be more frequent in small-sized seeds, while elliptic shape is more frequent in larger seeds [18]. In the Malvaceae, cardioid type seeds are associated with small herbs of annual cycle [29].

### 2.2. Seed Morphology in the Vitaceae

The seeds of the Vitaceae share structural characteristics. The endosperm presents in transversal section a typical “M” shape coincident with a pair of ventral in-folds and a dorsal chalaza knot allowing the identification of fossil seeds in this family [30].

For the application of morphology in taxonomy, characters of the seeds are selected and compared between different taxonomic groups. Frequently, the data concern measurements of defined structural components and the distances between well referenced seed positions. The results depend on the characters selected and the method used for comparison. Chen and Manchester applied successive PCA based on 57 characters to 252 seeds representative of 238 species belonging to 15 genera [31]. The conclusions were: (1) The seeds of *Leea*, *Cissus*, *Cyphostemma*, *Tetrastigma*, *Rhoicissus*, and *Cayratia* have a long or linear chalaza, visible from the ventral side and terminated very near the beak at the dorsal side, a condition that was termed ‘‘perichalaza’’ [32], whereas the seeds in the rest of the family usually have an oval chalaza, central in the dorsal position; (2) *Tetrastigma* and species of *Rhoicissus* have peculiar characteristics in their linear chalaza, which is located near the apical notch and extends their beak; their long, narrow, sometimes divergent ventral in-folds and their rugose surface. The authors conclude that, by comparison of selected sets of characters, the seeds can be distinguished to the generic level [31].

Seed shape in the Vitaceae is variable and seeds resembling geometric figures are frequent in this family. The seeds of *Ampelocissus* are pyriform, oval, or round in dorsal or ventral view [30]. The seeds of *Cissus* species are often described as globose with a pointed base, elliptic in outline or oblong (see, for example, [33,34,35]). These adjectives and other like sub-globose or terete are also applied to seed descriptions in other genera suggesting two important points: (1) The seeds of the Vitaceae are suitable for the comparison with geometric figures used as models. (2) The comparison may be quantitative, yielding measures that contribute to taxonomy. Table 2 contains a list of 131 species in the Vitaceae whose seeds have been observed for this work.

Seven geometric models for the description and quantification of seed shape in the Vitaceae were described before [45]. Models 1 to 5 were derived from modifications in the equations of the heart curve [63], Model 6 was derived from the pyriform curve [64], and Model 7 was obtained by the modification of two independent equations representing an ellipse, searching for the similarity with the seeds of cultivars of *V. vinifera*. In subsequent work, Model 7 revealed particularly useful, because models derived from it adjusted well to the shape of many cultivars of *V. vinifera* in the Spanish collection at IMIDRA [65]. The following section contains a description of new models obtained for this work and their examples in the Vitaceae. The first part presents geometric models shared by the Arecaceae and the Vitaceae, while the second part contains the description of new models based on a series of equations derived from the equation of an ellipse that apply to species in the Vitaceae.

## 3. Geometric Models for Seed Description and Quantification in the Vitaceae

### 3.1. Models Shared with the Arecaceae

The comparison of well-oriented seeds with geometric figures reveals the diversity of shapes in a family, adds precision to the description and permits quantification of seed shape. The geometric analysis of seed shape in the Vitaceae shows a variety of forms shared with the Arecaceae, such as lenses, superellipses and water drops [13], while their combination is not frequent in other plant families.

Lenses and superellipses [13] can be derived from the same formula:$${\left|\frac{x}{a}\right|}^{p}+{\left|\frac{y}{b}\right|}^{q}=1$$
with p,q>2 for superellipses and p>2, 1<q<2 for lenses [66,67]; (see Data Availability Statement section).

Figure 1 presents examples of seeds whose images resemble lenses of different length/width ratios: *Cissus sterculiifolia* [31], *Tetrastigma petraeum* [51] and *Cissus quadrangularis* [44].

The seeds of *Cissus reniformis*, *Cyphostemma laza* and *Ampelocissus bravoi* adjust to superellipses of different proportions (not shown). Additionally, in some *Tetrastigma* species the seeds resemble superellipses, for example: *Tetrastigma henryi* [52], *Tetrastigma campylocarpum* [51] and *T. caudatum* [51] (Figure 2).

Water drops and lemniscates described well the seeds of some species in the Arecaceae [13]. The former adapt well to the bi-dimensional shape of well-oriented images in the Vitaceae [45]. Figure 3 shows examples of seeds resembling elongated water drops, see [45] for quantitative measurements in *Cissus verticillata*.

The equation describing the water drop is given in [13,45], while the equation describing the lemniscate is given in [13].

### 3.2. Geometric Models for the Vitaceae: New Models Obtained from a Series of Equations Derived from an Ellipse

A family of equations will be described that simplify the design of new models according to the variety of seed types found in the Vitaceae [45]. The task of finding a model adapted to a new shape will be easier knowing which term of an existing equation may give the changes needed to obtain a given figure. With this objective, the equations corresponding to all models described in this section derive from the ellipse of equation:(1)$$1-x^{2}-b^{2}y^{2}=0$$
which can be expressed as:(2)$$\left(\sqrt{1-x^{2}}-b\;y\right)\left(\sqrt{1-x^{2}}+b\;y\right)=0$$
to remark the two explicit equations corresponding to the respective semi-ellipses that integrate it. Modifications in one of the factors, or in both, give equations of increasing complexity, whose graphic representations result in a variety of models.

A Water drop curve is obtained by the modification of Equation (2) to give:(3)$$\left(\sqrt{1-x^{2}}+b\;y\right)\left(\sqrt{1-x^{2}}+\frac{a}{50x^{2}+c}-b\;y\right)=0$$
with a=b=1, c=2. While the semi-ellipse corresponding to the left factor has not changed, the factor on the right determines the prominent part of the drop (beak). Increasing the value of a, increases the size of the beak (See Data Availability Statement section).

#### 3.2.1. Water Drop Models

The three models represented in Figure 4 result from changing the values of a,b and c and modifying other terms in Equation (2): Model VAM1 (a=0.6;b=1; c=2)  adjusts well to *Vitis amurensis*, *V. labrusca*, *V. rupestris* and *Cissus granulosa*; Model AGL1, a rounded Water drop, (a=0.3;b=1.1; c=1.6)  adjusts well to seeds of *Ampelopsis glandulosa*, *Tetrastigma triphyllum* and *Cissus fuliginea*; Model AAR1, an elongated Water drop, (a=3;b=1; c=5)  adjusts to seeds of *Ampelopsis arborea*, *Cissus campestris* and *C. willardii* (Data Availability Statement section). A slightly narrowed model adjusts well to seeds of *Tetrastigma hensleyanum* (not shown).

#### 3.2.2. Heart Curves

Simultaneous modifications in both terms of Equation (2) result in a variety of heart curves. Heart curves are obtained with selected values of a, b, c in:(4)$$\left(\sqrt{1-x^{2}}-\frac{a}{54x^{2}+9\left|x\right|+3}+y\right)\left(\sqrt{1-x^{2}}+\frac{b}{54x^{2}+9\left|x\right|+3}-y\right)=0$$

For example, Model ACO1 (see Figure 5) resulted from Equation (4) with a=1/2; b=1/3. Increasing *a*, reduces the size of the lower entry; increasing *b*, reduces the upper beak.

Model PHI1 (see Figure 5) was obtained by the following modification in Equation (4):(5)$$\left(\sqrt{1-x^{2}}+\frac{a}{10x^{2}+1}-y\right)\left(\sqrt{1-x^{2}}-\frac{b}{54x^{2}+9\left|x\right|+3}-y\right)=0$$
with *a* = 1/3; *b* = 1/2.

Model PPE1 resulted from:(6)$$\left(\sqrt{1-x^{2}}+\frac{a}{5x^{4}+25x^{2}+1}-\frac{9}{10}y\right)\left(\sqrt{1-x^{2}}-\frac{b}{5x^{4}+25{\left|x\right|}^{3}+1}-\frac{9}{10}y\right)=0$$
with *a* = *b* = 1/3. (Data Availability Statement section).

Changes in the above equations give modified water drop and heart curves. For example, changes in Equation (3) result in broadened heart curves, giving (i) Models ARO1, AJA1 and AER1 that describe the seeds of *Ampelocissus robinsonii, A. javalensis A. erdvendbergiana*, respectively; and (ii) Models AGR1 and ADE1 that resemble the seeds of *Ampelopsis grossdentata* and *A. denudata* (AGR1) and *A. delavayana* and *A. cantoniensis* (ADE1) respectively, see Figure 6 and Data Availability Statement section. The significance of differences between apparently similar models, such as PPE1 and AGR1, can be tested quantitatively on samples with a sufficient number of seeds. In principle, the difference at the basis of the figures (more flat and with a plane entry in PPE1) justifies the separation of the two models.

Departing from the models described, it is possible to find new figures specific for seeds in other species; for example, seeds of *Ampelocissus cavicaulis*, *A. macrocirrha,* and *A. ochracea* share with *A. javalensis* the Squared Heart Curve (SqHC) type related to Model AJA1. Other models, such as AGR2, may fit better the shape of seeds of *A. grantii* and *A. latifolia* (Figure 7).

#### 3.2.3. Pear Curves and Other Elongated Models

Some seed images resemble water drops in their overall shape; nevertheless, they show broader basis than waterdrops. Figure 8 shows the models YAU1, COL1, VAE1 and AME1 that correspond respectively to the shapes of *Yua austro-orientalis* and *Cissus trianae* (YAU1), *Cayratia oligocarpa* (COL1), *Yua chinensis* and *Vitis aestivalis* (VAE1) and *Ampelopsis megalophylla* (AME1). These four models share similar values of aspect ratio that justify the inclusion of model COL1 here with preference to the Squared Heart Curve (SqHC) group. Seeds of some species of *Ampelocissus* (e.g., *A. acapulcensis, A. bombycina, A. bravoi* can fit either model COL1 or models derived from it (not shown)).

The seeds of *Rhoicissus rhomboidea* and of many species of *Tetrastigma* resemble polarized ellipses with an end bi-lobulated and the other acute. We have termed this morphological group as the Elongated Superellipse-Heart Curve (ESHC) (Figure 9).

### 3.3. A Summary of Geometric Types in Seeds of the Vitaceae

Table 3 contains a summary of the morphological types for the Vitaceae according to the similarity of the seeds with geometrical figures. Ten morphological groups are described, three being present also in the Arecaceae [13] (lenses, superellipses and elongated water drops; termed respectively G I, G II and G III in Figure 10), three additional groups were described before for the Vitaceae [45] (water drops, heart curves and elongated heart curves; named G IV, G V and G VI in Figure 10), and four new groups are based on new models original from this work (G VII to G X). Three of the later models are particular for some genera and species. These are: (1) Heart curves of the *Cayratia* and *Pseudocayratia* types, with a marked entry at the basis and an acute protuberance (G VIII in Figure 10); (2) heart curves of the Squared Heart Curves (SqHCs) type in *Ampelocissus* and broadened models of *Ampelocissus* and *Ampelopsis* (G IX in Figure 10), and (3) Elongated Superellipse-Heart Curves (ESHCs), frequent in *Tetrastigma* species and observed also in *Cissus* species and *R. rhomboidea* (G X in Figure 10).

In general, the distribution of morphological types is not in close agreement with the current taxonomic classification; nevertheless, some results may be summarized in this aspect. First, the seeds of the Elongated Superellipse-Heart Curves (ESHCs) type (Group X) are more frequent in *Tetrastigma* and have been observed in *Rhoicissus* and *Cissus*, but not in species of other genera. While many seeds in species of *Ampelopsis, Parthenocissus* and *Vitis* share the typical shapes of water drop and heart curves, the squared heart curve (SqHC) type (Group IX) has been predominantly observed in *Ampelocissus* and *Ampelopsis*. A number of species remain undefined due to one of these two reasons: First, their irregular seed shape making difficult the identification of an adequate model (*Cayratia geniculata*, *Cissus antarctica*) and, second, the seed images have geometric shapes but the identification of the model with the corresponding equation is pending (*Tetrastigma delavayi*, *T. rumicispermum*). In addition, further work will be done on the seeds of *Vitis* species.

## 4. Discussion

Morphology has not received the attention due in recent decades due to the increased emphasis on molecular approaches, but the importance of descriptive aspects is rising [68]. Seed morphology in particular may provide the basis for developments in Ecology and Evolution. The morphological analysis shows a similarity in seed shape between two families from the Core Angiosperms that are not related traditionally by taxonomic criteria: the Arecaceae and the Vitaceae. Both families belong to very different clades, the Vitaceae to the Eudicot clade and the Arecaceae to the Monocot clade [1,6,69,70], and although the embryos of the former have two cotyledons and the latter have only one, their similarities in seed shape are in agreement with both families having endospermic seeds [3,4,70]. The seeds of the Arecaceae and the Vitaceae present a great diversity, including a combination of forms relatively infrequent in other plant families. Ellipses, ovals and cardioids are geometric forms frequent in plant families [18]; in contrast, superellipses are not so frequent. These adjust better to intermediate shapes between ellipses and rectangles. Other figures shared by the seeds of the Vitaceae and the Arecaceae are lenses and water drops of diverse proportions.

A distinctive aspect of seed morphology in the Vitaceae is the adjustment of their seeds to a diversity of water drops, heart curves and related figures. A series of variations derived from the equation of an ellipse have been described; their graphical representations give water drops and heart curves resembling with precision the seeds of diverse species of the Vitaceae. Both types of figures can be described as products from the modification of ellipses to obtain one pole acute and the other rounded or bi-lobulated.

The new models here described define the seed silhouettes of species in the diverse subfamilies of the Vitaceae. In addition to three groups based on models shared with the Arecaceae, and three other groups described before (water drops, heart curves, elongated heart curves) [45], four new groups have been described. One of them corresponds to other types of elongated curves, and the remaining three are more specific. These correspond to: (1) Heart curves of the *Cayratia* and *Pseudocayratia* types, with a marked entry at the basis and an acute protuberance; (2) heart curves of the Squared Heart Curves (SqHCs) type in *Ampelocissus* and broadened models of *Ampelocissus* and *Ampelopsis*, and (3) Elongated Superellipse-Heart Curves (ESHCs), frequent in *Tetrastigma* species and observed also in *Cissus* species and *R. rhomboidea*.

The importance of seed morphology in taxonomy has been described for a long time, see, for example, [71]. Nevertheless, not all characters of seed morphology have the same relevance, and, in many instances, the lack of a morphological diagnostic key may be due to a high degree of homoplasy [72,73]. The similarity of the seed silhouette to a geometric model is the result of a complex process of development, and thus less submitted to homoplasy; in consequence, it may be a good character in taxonomy. In addition, the visualization of geometric figures that share the form of seeds may contribute to their classification complementing the results of artificial vision techniques [74,75,76].

In addition to taxonomy, classification based on seed shape acquires relevance in other research areas. Members of both families, the Arecaceae and the Vitaceae, were present in the Neotropical flora in the Eocene [14], and their fruits have been in the human diet for a long time [75,76]. Additionally, both families have been studied by means of phytoliths, microfossils useful in archaeobotany and archaeology [77,78].

In the first paragraph of the introduction to his book entitled *L’Évolution créatrice*, Henri Bergson recognized the importance of Geometry stating that *nôtre intelligence triomphe dans la géometrie, où se révelè la parenté de la pensée logique avec la matière inerte* [79]. Unfortunately, in the following pages of this text, the author abandoned the study of geometry, a model of precision, to enter the rhetorics of evolution. An approach to seed geometry in palms and grapes could support the words of P.B. Tomlinson (1990): “Palms are not then merely emblematic of the tropics, they are emblematic of how the structural biology of plants must be understood before evolutionary scenarios can be reconstructed” [80], quoted in [69].

## 5. Conclusions

Ten morphological types are described in the Vitaceae. Seven of them are general and three specific. Among the general types, three are shared with the Arecaceae and correspond to geometric figures well described (lenses, superellipses and elongated waterdrops). Four additional groups include waterdrops, normal or rounded, heart curves, normal or rounded, elongated heart curves and other elongated curves, respectively. Finally, the three specific types correspond to heart curves of the *Cayratia* and *Pseudocayratia* types, heart curves of the Squared Heart Curve (SqHC) type of *Ampelocissus* and *Ampelopsis*, and Elongated Superellipse-Heart Curves (ESHCs), frequent in *Tetrastigma* species and observed also in *Cissus* species and *R. rhomboidea*. All these groups are defined by geometric models obtained by the representation of algebraic equations. Modifications in the equations result in models adjusting to the shape of seeds for each species.

## Figures and Tables

**Figure 1 plants-10-01695-f001:**
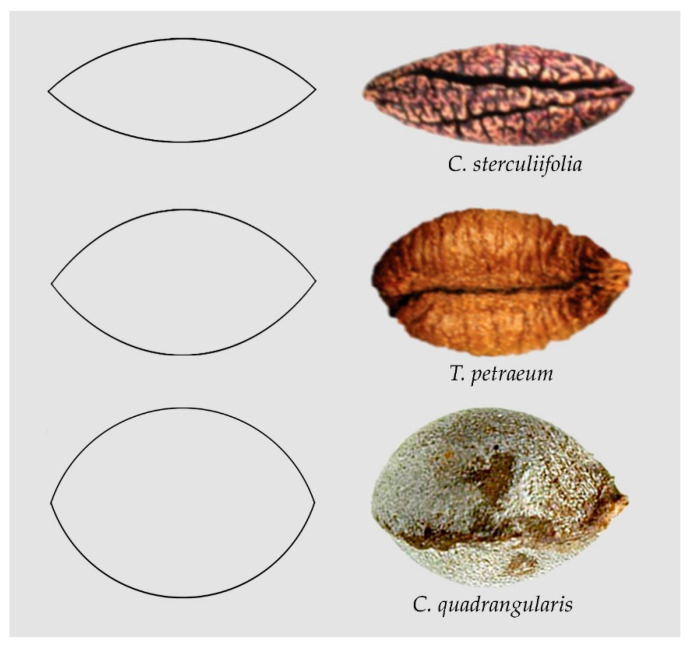
The images of seeds of *Cissus sterculiifolia* [31], *Tetrastigma petraeum* [51] and *Cissus quadrangularis* [44] resemble lenses of different proportions.

**Figure 2 plants-10-01695-f002:**
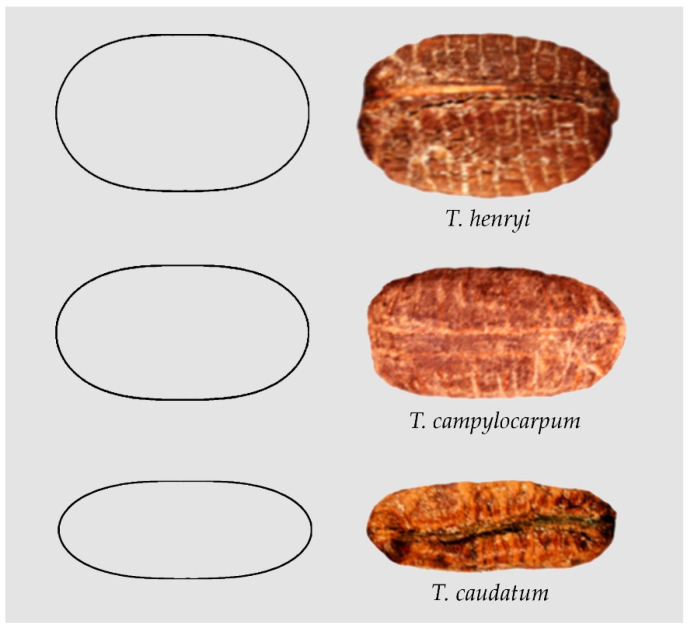
Examples of seed images resembling superellipses with different length/width ratios: First row: *Tetrastigma henryi* [51]. Second row: *Tetrastigma campylocarpum* [51]. Third row: *T. caudatum* [51].

**Figure 3 plants-10-01695-f003:**
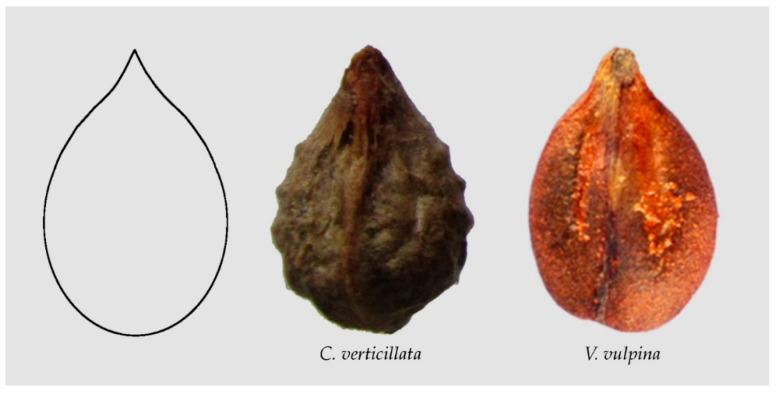
Examples of seed images resembling elongated water drops. From left to right: The model for an elongated water drop (Model 4 in [45]), seeds of *Cissus verticillata* and *Vitis vulpina*.

**Figure 4 plants-10-01695-f004:**
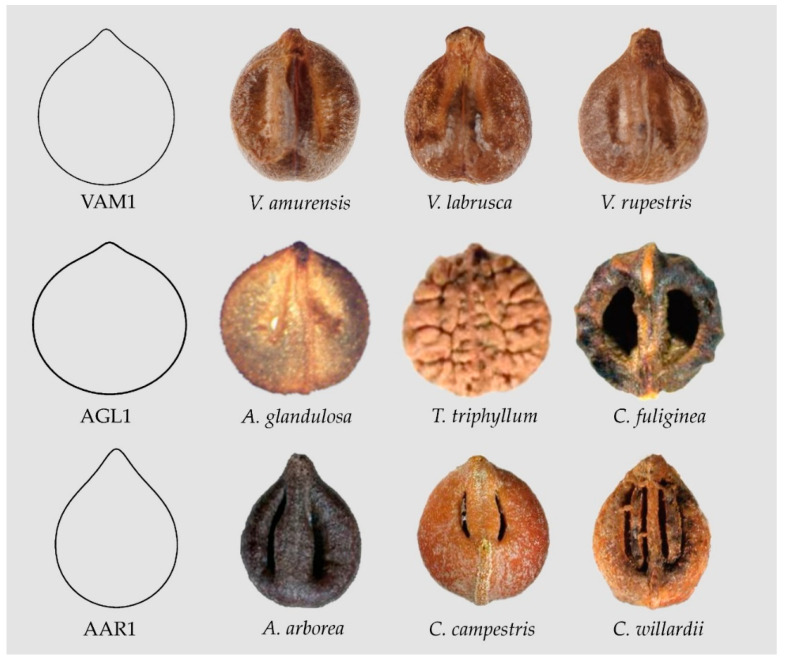
Seed images resembling water drops. From top to bottom: Model VAM1 (*Vitis amurensis*, *V. labrusca*, *V. rupestris*), Model AGL1 (*Ampelopsis glandulosa*, *Tetrastigma triphyllum*, *Cissus fuliginea*), Model AAR1 (*Ampelopsis arborea*, *Cissus campestris*, *C. willardii*).

**Figure 5 plants-10-01695-f005:**
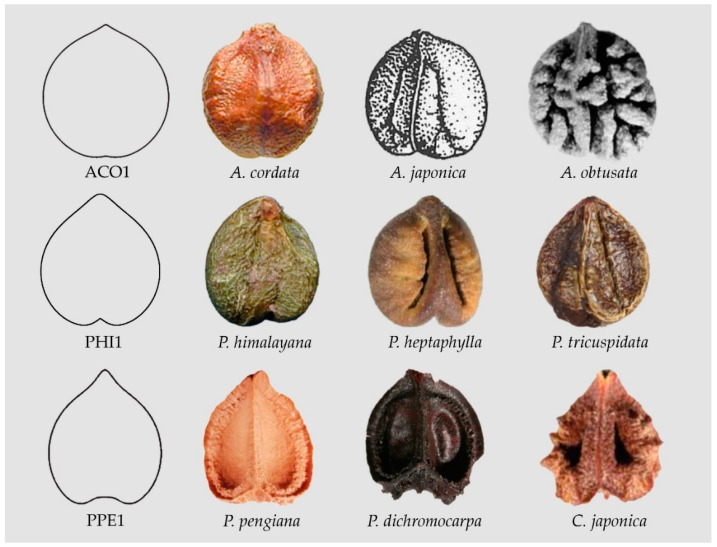
Seeds resembling heart curves. From top to bottom: Model ACO1 (*Ampelopsis cordata*, *A. japonica*, *A. obtusata*), Model PHI1 (*Parthenocissus himalayana*, *P. heptaphylla*, *P. tricuspidata*), Model PPE1 (*Pseudocayratia pengiana*, *P. dichromocarpa*, *Cayratia japonica*).

**Figure 6 plants-10-01695-f006:**
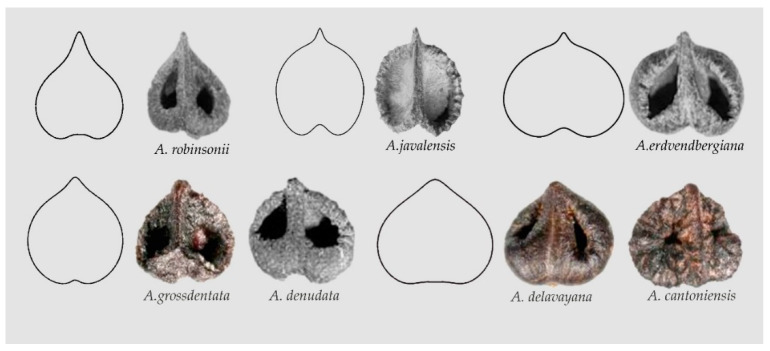
Models of broadened heart curves and seeds resembling each of them in *Ampelocissus* and *Ampelopsis*. (**Top**): Model ARO1, *Ampelocissus robinsonii*, Model AJA1, *Ampelocissus javalensis*, Model AER1, *A. erdvendbergiana*. (**Bottom**): Model AGR1, *Ampelopsis grossedentata*, *A. denudata*; Model ADE1, *A. delavayana*, *A. cantoniensis*.

**Figure 7 plants-10-01695-f007:**
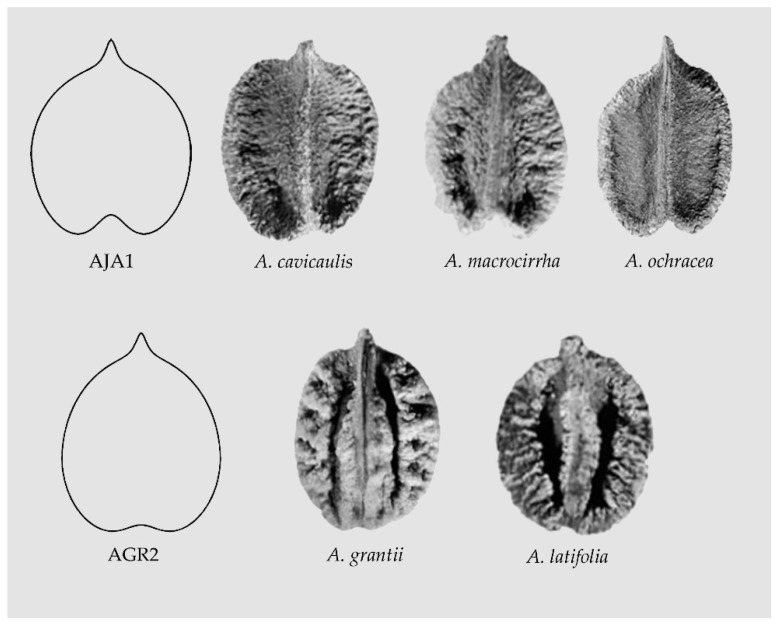
The seeds of many species of *Ampelocissus* are related with the Squared Heart Curve (SqHC). Model AJA1 adapts well to *Ampelocissus cavicaulis*, *A. macrocirrha* and *A. ochracea.* Model AGR2 represents better the shape of seeds of *A. grantii* and *A. latifolia*.

**Figure 8 plants-10-01695-f008:**
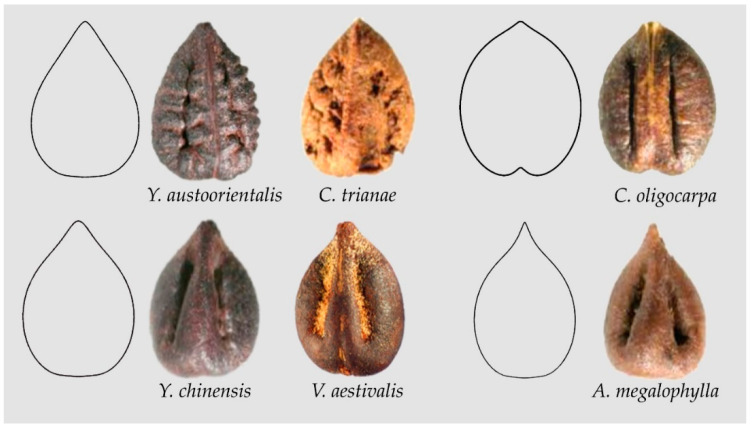
Models YAU1, COL1, VAE1 and AME1 with their respective seeds. *Yua austro-orientalis* and *Cissus trianae* (YAU1), *Cayratia oligocarpa* (COL1), *Yua chinensis* and *Vitis aestivalis* (VAE1) and *Ampelopsis megalophylla* (AME1).

**Figure 9 plants-10-01695-f009:**
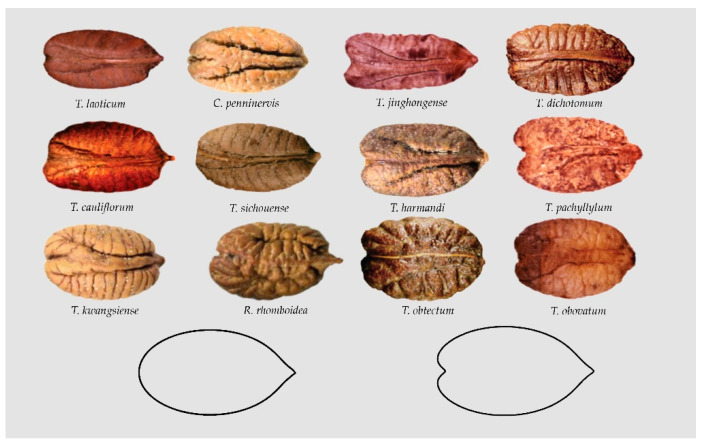
Seeds of *Rhoicissus rhomboidea* and many species of *Tetrastigma* resemble polarized ellipses with a side rounded or bi-lobulated and the other acute.

**Figure 10 plants-10-01695-f010:**
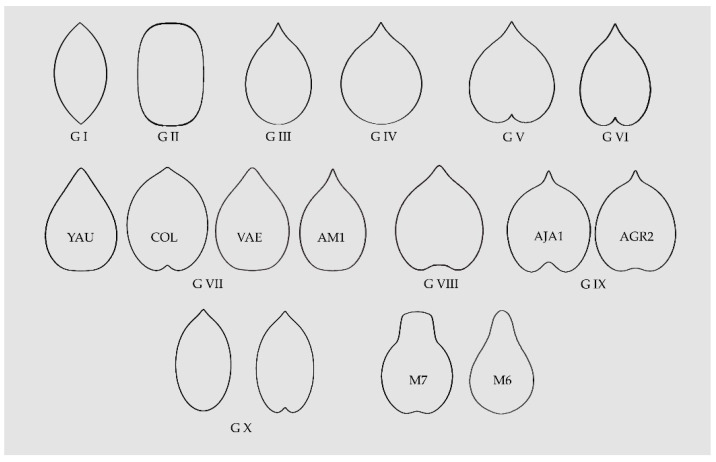
A summary of the models found for the description and quantification of seed shape in the Vitaceae. G I, G II and G III are lenses, superellipses and elongated water drops, respectively; G IV, G V and G VI correspond to water drops, heart curves and elongated heart curves, respectively; G VII contains four models corresponding to other elongated curves; G VIII presents an example of the heart curves of the *Cayratia* and *Pseudocayratia* types; G IX, heart curves of the Squared Heart Curves (SqHCs) type in *Ampelocissus* and broadened models of *Ampelocissus* and *Ampelopsis,* and G X, Elongated Superellipse-Heart Curves (ESHCs), frequent in *Tetrastigma* species and observed also in *Cissus* species and *R. rhomboidea.* Labelled as M7 and M6 are two models used in the description of seeds of grape varieties and as precursors for other models [45,65].

**Table 1 plants-10-01695-t001:** A summary of the taxonomy of the Vitaceae. The approximate number of species in each tribe and genus is given between parentheses. Data adapted from [3].

Tribe	Genera
I. Ampelopsideae (47)	*Ampelopsis* Michx. (18) *Nekemias* Raf. (9)
	*Rhoicissus* Planch. (14)
	*Clematicissus* Planch. (6)
II. Cisseae (300)	*Cissus* L. (300)
III. Cayratieae (368)	*Cayratia* Juss. (25)
	*Causonis* Raf. (30)
	*Acareosperma* Gagnep. (1)
	*Afrocayratia* (7)
	*Cyphostemma* (Planch.) Alston (200)
	*Pseudocayratia* J. Wen, L.M.Lu and Z.D. Chen (5)
	*Tetrastigma* (Miq.) Planch. (100)
IV. Parthenocisseae (16)	*Parthenocissus* Planch. (14)
	*Yua* C.L.Li (2)
V. Viteae (190)	*Ampelocissus* Planch. (115)
	*Vitis* L. (75)

**Table 2 plants-10-01695-t002:** A summary of the 131 species for which seed shape has been analysed in this work.

Tribe (Species Observed/Total)	Genera (Species Observed/Total)	Species (References for the Images)
I. Ampelopsideae (15/47)	*Ampelopsis* Michx. (13/18)	*Ampelopsis aconitifolia* [36], *A. arborea* [37], *A. bodinieri* [36], *A. cantoniensis* [31,36], *A. cordata* [38], *A. chaffanjoni* [36], *A. delavayana* [31], *A. denudata* [30], *A. glandulosa* [39], *A. grossedentata* [31], *A. humulifolia* [36], *A. japonica* [36], *A. megalophylla* [31,36]
	*Rhoicissus* Planch. (2/14)	*Rhoicissus revoilii* [31], *R. rhomboidea* [31]
II. Cisseae (33/300)	*Cissus* L. (33/300)	*Cissus antarctica* [31], *C. aralioides* [35,40], *C. barbeyana* [40], *C. bosseri* [40], *C. cactiformis* [40], *C. campestris* [31,41,42], *C. cornifolia* [40], *C. descoingsii,* [31,41], *C. diffusiflora* [40], *C. elongata* [40], *C. erosa* [43], *C. floribunda* [40], *C. fuliginea* [31], *C. granulosa* [31], *C. hastata* [40], *C. hypoglauca* [31], *C. integrifolia* [40,42], *C. leucophlea* [40], *C. penninervis* [31], *C. petiolata* [40], *C. pileata* [40], *C. populnea* [40], *C. quadrangularis* [44], *C. reniformis* [31,41], *C. repens* [40], *C. sciaphila* [40], *C. smithiana* [40], *C. sterculiifolia* [31], *C. subtetragona* [40], *C. trianae* [31], *C. tuberosa* [42], *C. verticillata* [31,41,42,45], *C. willardii* [42],
III. Cayratieae (40/365)	*Causonis* Raf. (1/9)	*Causonis* sp. [46]
	*Cayratia* Juss. (7/60)	*Cayratia cheniana* [46], *C. geniculata* [31], *C. imerinensis* [47], *C. japonica* [31,48], *C. oligocarpa* [31], *C. saponaria* [31], *C. sp*. [African, [46]]
	*Cyphostemma* (Planch.) Alston (3/200)	*Cyphostemma elephantopus* [49], *C. laza* [31], *C. junceum* [31]
	*Pseudocayratia* J. Wen, L.M.Lu and Z.D.Chen (3/5)	*Pseudocayratia dichromocarpa* [50], *P. pengiana* [50], *P. speciosa* [50,51]
	*Tetrastigma* (Miq.) Planch. (26/100)	*Tetrastigma campylocarpum* [52], *T. cauliflorum* [52], *T. caudatum* [52], *T. delavayi* [52], *T. dichotomum* [51], *T. formosanum* [52], *T. harmandi* [31], *T. hemsleyanum* [31,52], *T. henryi* [52], T. *hypoglaucum* [52], *T. jinghongense* [52], *T. kwangsiense* [30,31], *T. lanceolarium* [30], *T. laoticum* [52], *T. obovatum* [51,52], *T. obtectum* [51,52], *T. pachyllylum* [52], *T. pedunculare* [31,51,52], *T. petraeum* [52], *T. retinervum* [52], *T. rumicispermum* [31,51,52], *T. serrulatum* [52], *T. sichouense* [52], *T. thorsborneorum* [52], *T. triphyllum* [31,52], *T. xishuangbannaense* [31,52]
IV. Parthenocisseae (11/16)	*Parthenocissus* Planch. (9/14)	*Parthenocissus dalzielii* [36], *P. heptaphylla* [31], *P. henryana* [36], *P. heterophylla* [36], *P. himalayana* [53], *P. laetevirens* [36], *P. quinquefolia* [54], *P. tricuspidata* [36,37,45], *P. vitacea* [31]
	*Yua* C.L.Li (2/2)	*Yua austro-orientalis* [31], *Y. chinensis* [31]
V. Viteae (32/190)	*Ampelocissus* Planch. (13/115)	*Ampelocissus acapulcensis* [30], *A. bombycina* [30], *A. bravoi* [42], *A. cavicaulis* [30], *A. erdvendbergiana* [30], *A. grantii* [30], *A. javalensis* [30,42], *A. latifolia* [30], *A. macrocirrha* [30], *A. martinii* [42], *A. obtusata* [30], *A. ochracea* [30], *A. robinsonii* [30]
	*Vitis* L. (19/75)	*Vitis aestivalis* [55], *V. amurensis* [45,56], *V. brandoniana* [54], *V. cinerea* [57], *V. eolabrusca* [54], *V. flexuosa* [54], *V. grayensis* [58], *V. labrusca* [45,54,59], *V. lanatoides* [58], *V. latisulcata* [58], *V. palmata* [60], *V. pseudorotundifolia* [54], *V. rostrata* [54], *V. rotundifolia* [31,41,54], *V. rupestris* [45], *V. tiliifolia* [61], *V. tsoi* [31,41], *V. vulpina* [59,62], *V. wilsoniae* [31,41]

**Table 3 plants-10-01695-t003:** A summary of groups based on morphological seed types for the analysed species in the Vitaceae. The number of cases found in each group is given between dashes.

Group (Geometric Model)	Examples
Group I (Lenses)-3-	*Cissus quadrangularis* [44], *C. sterculiifolia* [31], *Tetrastigma petraeum* [52]
Group II (Superellipses)-7-	*Ampelocissus bravoi* [42], *C. reniformis* [31,41], *Cyphostemma elephantopus* [49], *C. laza* [31], *Tetrastigma campylocarpum* [52], *T. caudatum* [52], *T. henryi* [52]
Group III (Elongated water drops)-15-	*Ampelopsis arborea* [37], *Cayratia imerinensis* [47], *Cissus aralioides* [35,40], *C. cornifolia* [40], *C. erosa* [43], *C. integrifolia* [40,42], *C. petiolata* [40], *C. pileata* [40], *C. populnea* [40], C. *verticillata* [31,41,42,45], *C. sciaphila* [40], *C. smithiana* [40], *C. willardii* [42], *Cyphostemma junceum* [31], *V. vulpina* [59,62]
Group IV (Water drops, normal or rounded)-14-	*Ampelopsis bodinieri* [36], *A. glandulosa* [36,39], *A. humulifolia* [36], *Cayratia cheniana* [46], *Cissus campestris* [31,41,42], *C. fuliginea* [31], *C. tuberosa* [42], *C. granulosa* [31], *Parthenocissus dalzielii* [36], *Tetrastigma triphyllum* [31,52], *Vitis amurensis* [45,56], *V. labrusca* [45,54,59], *V. palmata* [60], *V. rupestris* [45]
Group V (Heart curves normal or rounded)-19-	*Ampelopsis aconitifolia* [36], *A. chaffanjoni* [36], *A. cordata* [38], *A. japonica* [36], *Parthenocissus heptaphylla* [31], *P. heterophylla* [36], *P. henryana* [36], *P. himalayana* [52,53], *P. quinquefolia* [54], *P. vitacea* [31], *P. tricuspidata* [36,37,45], *Rhoicissus revoilii* [31], *T. lanceolarium* [30], *Vitis cinerea* [57], *V. flexuosa* [54], *V. lanatoides* [58], *V. latisulcata* [58], *V. tsoi* [31,41], *V. wilsoniae* [31,41]
Group VI (Elongated Heart curves)-6-	*Ampelocissus acapulcensis* [30], *Cissus oligocarpa* [31]. *V. eolabrusca* [54], *V. grayensis* [58], *V. pseudorotundifolia* [54], *V. tiliifolia* [61]
Group VII (Other elongated types)-11-	*Ampelopsis megalophylla* [31,36], *Causonis* sp. [46], *Cayratia saponaria* [31], *Cissus trianae* [31], *C. hypoglauca* [31], *Parthenocissus laetevirens* [36], *T. hypoglaucum* [52] *Vitis aestivalis* [55], *V. rotundifolia* [31,41,54], *Yua austro-orientalis* [31], *Y. chinensis* [31]
Group VIII (Heart curves of the *Cayratia* and *Pseudocayratia* types)-7-	*Cayratia japonica* [31,48], *Cayratia* sp. [African, [46]], *Pseudocayratia dichromocarpa* [50], *P. pengiana* [50], *P. speciosa* [50,52], *Tetrastigma formosanum* [51], *T. pedunculare* [31,51,52]
Group IX (Heart curves of the SqHC type of *Ampelocissus* and *Ampelopsis*)-15-	*Ampelocissus bombycina* [30], *A. cavicaulis* [30], *A. erdvendbergiana* [30], *A. grantii* [30], *A. javalensis* [30,42], *A. latifolia* [30], *A. macrocirrha* [30], *A. martinii* [42], *A. obtusata* [30], *A. ochracea* [30], *A. robinsonii* [30], *Ampelopsis cantoniensis* [31,36], *A. delavayana* [31], *A. denudata* [30], *A. grossedentata* [31]
Group X Elongated Superellipse-heart curves-16-	*Cissus elongata* [40], *C. penninervis* [31], *Rhoicissus rhomboidea* [31], *Tetrastigma hemsleyanum* [31,52], *T. jinghongense* [52], *T. laoticum* [52], *T. cauliflorum* [52], *T. dichotomum* [51], *T. harmandi* [31], *T. pachyllylum* [52], *T. kwangsiense* [30,31], *T. obovatum* [51,52], *T. obtectum* [51,52], *T. retinervum* [51], *T. serrulatum* [52], *T. sichouense* [52]
Undefined-18-	*Cayratia geniculata* [31], *Cissus antarctica* [31], *C. barbeyana* [40], *C. bosseri* [40], *C. cactiformis* [40], *C. descoingsii,* [31,41], *C. diffusiflora* [40], *C. floribunda* [40], *C. hastata* [40], *C. leucophlea* [40], *C. repens* [40], *C. subtetragona* [40], *T. delavayi* [52], *T. rumicispermum* [31,51,52], T. *thorsborneorum* [52], *T. xishuangbannaense* [31,52], *V. brandoniana* [54], *V. rostrata* [54]

## Data Availability

The Mathematica code for Geometric Models is given in: https://zenodo.org/record/4942111#.YMbkrfkzaM8.

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
