# Peer review of "Seed Geometry in the Vitaceae"

_plants, 2021, doi:10.3390/plants10081695_

Round 1

Reviewer 1 Report

In an attached document, I suggest a list of revisions in the text and questions about the manuscript

The main suggestion is the inclusion of a complete figure with all the variability of models of seeds in the family Vitaceae, including models of previous papers.

Author Response

Dear Evaluator,

Thank you very much for reviewing our article. Your commentaries and questions have contributed notably to improve it. All the corrections indicated have been introduced in the text, and responses to the questions are given in the actual version.

The question concerning the differences between models PPE1 and AGR1 is now commented on lines 240-244:

“The significance of differences between apparently similar models, such as PPE1 and AGR1, can be tested quantitatively on samples with a number of seeds. In principle, the difference at the basis of the figures (more flat, and with a plane entry in PPE1) justifies the separation of the two models.”

The reason for not having included model COL1 in the group of squared heart curves is based on similar values of aspect ratio in the models belonging to each of the groups as explained now on lines 266-268:

“These four models share similar values of aspect ratio that justify the inclusion of model COL1 in this group with preference to the Squared Heart Curve (SqHC).”

The reasons for keeping some species as undefined types in table 3 are now given (lines 317-321).

“A number of species remain undefined due to one of these two reasons: First, their irregular seed shape making difficult the identification of an adequate model (Cayratia geniculata, Cissus antarctica), and, second, the seed images have geometric shapes but the identification of the model with the corresponding equation is pending (Tetrastigma delavayi, T. rumicispermum). In addition, further work will be done on the seeds of Vitis species.”

We appreciate your suggestion of adding a complete figure with all the variability of models of seeds in the family Vitaceae, including models of previous papers. Figure 10, the corresponding legend and explanation in the text have been added following your indication. This helps to make clear the morphological types in seeds of this family.

Thanking you for the careful review,

Emilio Cervantes

Reviewer 2 Report

The manuscript entitled "Seed Geometry in the Vitaceae" is important because it brings order to the great variability of the Vitaceae.

The use of Seed Geometry or in any case of computer vision, applied to botanical sciences, allows to optimize the times for identification at a specific level. 

Author Response

Dear Reviewer,

Thank you very much for your revision of our article. We are glad that you find the article suitable for publication.

Thanking you for the review,

Emilio Cervantes

Reviewer 3 Report

Dear Authors,

The subject of the study is interesting and topical, with high scientific and practical importance.

The introduction is in accordance with the subject and correctly presented. Numerous scientific articles of recent date and in concordance to the topic of the study were consulted.

Methodology of the study was clearly presented, under the conditions of a review article.

The communicated results have been analyzed and interpreted in accordance with the current methodology.

The discussions are appropriate, in the context of the results, and was conducted compared to other studies in the field.

The scientific literature, to which the reporting was made, is recent and representative in the field.

However, the review of the article revealed some minor issues, which were noted in the article.

1.

Small differences in the length of some text lines.

eg

Rows 29 - 41, 45 - 57 compared to the following.

A check is recommended, according to Instructions for Authors, Microsoft Word template, Plants journal.

2

The subchapters of the article are recommended to be in accordance with the settings presented in Instructions for Authors, Plants journal.

3.

Some equations are not numbered.

eg

Page 10, rows 225, 228

4.

In the conditions of a Review article, it is recommended to indicate the bibliographic sources for the presented figures, as appropriate.

5.

References

The entire References chapter should be revised, in accordance with the Instructions for Authors, Plants Journal.

According to Instructions for Authors, Plants journal

"References should be described as follows, depending on the type of work:

Journal Articles:

  1. Author 1, A.B.; Author 2, C.D. Title of the article. Abbreviated Journal Name Year, Volume, page range."

Sample:

“Ridsdale, C.E. A revision of the family Leeaceae. Blumea 1974, 22, 57-100.”

instead of:

“Ridsdale, C. E. A revision of the family Leeaceae. Blumea 1974, 22, 57-100.”

no space between the initial of the author's name "M." and ";"

eg

Page 8, row 392:

"Rossetto, M.;" instead of "Rossetto, M. ;"

Instructions for Authors, Plants journal, does not recommend that every word in the title of the article be capitalized.

“Cervantes, E.; Martín-Gómez, J.J. Seed shape description and quantification by comparison with geometric models. Horticulturae 2019, 5, 60, https://doi.org/10.3390/horticulturae5030060”

instead of:

“Cervantes, E.; Martín-Gómez, J.J. Seed Shape Description and Quantification by Comparison with Geometric Models. Horticulturae 2019, 5, 60, https://doi.org/10.3390/horticulturae5030060”

No space between the initials of the authors' names

Abbreviated Journal Name

Year - Font style: bold

Volume – Font style: italic

It is recommended to check and correct, as appropriate, the entire chapter of references, in accordance with Instructions for Authors, and Microsoft Word template, Plants journal.

Author Response

Dear Reviewer,

Thank you very much for your revision of our article. Your commentaries and questions have contributed notably to improve it. All the corrections indicated have been introduced in the text.

The sub-chapters of the article are now in accordance with the Instructions for Authors. All the equations have been numbered as indicated. Bibliographic sources for the images presented in the figures are now given in the Supplementary Materials section. All the references in the reference list have been revised and corrected according to your indications.

Thanking you very much for the careful review,

Emilio Cervantes

Reviewer 4 Report

The manuscript is well written and comes with a very nice and informative collection of pictures of seeds in the Vitaceae. One can feel the hard work behind this manuscript. However, it is hard to depict the actual value of the study presented here. Why is it needed a framework for the description of seed morphology.

It is not clear whether the manuscript is a review of a research where the authors develop models for the description of the seed shape of Vitaceae (not sure that is the purpose of those models). This indicates that the introduction is not written in such a way that any reader could understand the purpose of the work,

Line 118- The authors concluded that, by comparison of selected sets of characters, the seeds can be distinguished to the generic level. ¿Is this a great finding? Flowers, leaves, and stalks are commonly used as traits to classify and to identify pants species, no wonder that seed of related species show related characteristics.

Seed show different shapes, it is right. Seed in the same genera are alike and they fit into similar mathematical models, right. Generally speaking, seeds attach to different mathematical models, right. But, so what? What are you contributing with to the knowledge of the species, other than describing this very important attribute of each species. Is this for the shake of the botanical description? If so, this is  big hammer for a tiny nut.

Is it possible to establish a phylogenetic relation among genera and their seed shape? Is there any evolutionary implication related to the seed shape? Or, is there any ecological implication of having pear or ovoidal seeds? These, or other relations would make of the study a more useful one. In its current form is merely descriptive and bring little advance in the area of seed ecology, or broadly speaking in Botany.

Throughout the entire manuscript there is a continuous reference to a previous paper on size and shape of seed in the Arecaceae family. In addition, there are comparisons between that family and Vitacea, but no where it is explained that the objective of the study implies such a comparison, for what I’m very confused on what the authors want to convey (sorry). This continues in the discussion. Well, do so, but indicate in the introduction that this is part of the study.

What is more, if your contribution is the development of the models, add a Material and Methods section indication how measurements of seeds were obtained, and how the models were developed.

The conclusion section is a mere summary of the content in the manuscript, not concluding anything with biological meaning.

Author Response

Dear Reviewer,

Thank you very much for your revision of our article.

A framework in seed morphology is actually needed by two reasons. First, as a part of a general framework in Morphology, a scientific discipline that has not received the due attention in recent decades as a consequence of the increased emphasis on molecular approaches. Second, the description of seed shape may be important due to the particular relevance of the seeds, structures characteristic of Angiosperms. This is now indicated at the beginning of the discussion section, and a new reference has been added to illustrate this important aspect.

The purpose of the work is indicated at the end of the introduction (rows 63-66):

“The main objective of this review is to provide a framework for the description of seed morphology in the Vitaceae based on geometric models. A recent review of this subject in the Arecaceae described morphological types in the seeds of this family based on the similarity of seed images to geometric figures, like ellipses, ovals and others [13].”

The third commentary refers to our mentioned conclusion of the work of Chen and Manchester. Yes, we think that is an important conclusion that the species may be distinguished by the shape of their seeds. It is a general conclusion, but it needs much more precision. The objective of our work is to provide the tools for this.

Botanical description is at the basis for the study of Ecology and Evolution, and as you mention in the commentary, seed shape is a very important attribute that we want to describe accurately. Descriptive analysis provides the basis for further studies in other fields, as those that you have indicated. This is expressed in the new paragraph at the beginning of the Discussion section.

Concerning your question on how the models were developed, this is described in the section 3.2. entitled 3.2. Geometric models for the Vitaceae: New models obtained from a series of equations derived from an ellipse. In consequence there is no need to add a Materials and Methods section, that in addition would be contradictory in a Review type article.

Thanking you for the review,

Emilio Cervantes
